# Investigation of the effects of probiotic, Bacillus subtilis on stress reactions in laying hens using infrared thermography

**Maria Soroko** [1]*, **Daniel Zaborski** [2]

**1** Department of Horse Breeding and Equestrian Studies, Institute of Animal Breeding, Wroclaw University of Environmental and Life Sciences, Wroclaw, Poland, **2** Department of Ruminants Science, West Pomeranian University of Technology, Szczecin, Poland

* kontakt@eqma.pl

## Abstract

The goal of the study was to assess whether tonic immobility (TI)-induced stress reactions in laying hens can be reduced by probiotic supplementation and if the changes in body surface temperature, as a stress indicator, are genetically dependent and can be detected using infrared thermography (IRT). Seventy-one white and 70 brown hens were used. Hens were randomly assigned to three treatments at 1-day-old: beak trimmed and fed a regular diet; non-beak trimmed and fed a regular diet; and non-beak trimmed and fed a diet supplemented with probiotics, Bacillus subtilis. At 40 weeks of age, hens were tested for TI reactions. Eye and face temperatures were measured with IRT immediately before and after TI testing. Results revealed that the probiotic supplementation did not affect hens' stress responses to TI testing; the left and right eye temperatures increased by 0.26s°C and 0.15°C, respectively, while right face temperature tended to increase following TI testing. However, the right eye (32.60°C for white, and 32.35°C for brown) and face (39.51°C for white, and 39.36°C for brown) temperatures differed significantly among genetic lines. There was a positive correlation between TI duration and the changes of the left and right eye temperatures after TI testing in white hens. Based on these results, hens experienced TI-induced surface temperature changes that were detectable using IRT. White hens experienced greater stress reactions in response to TI than brown hens. However, supplementation with Bacillus subtilis did not attenuate hens' reaction to TI testing.

## Introduction

Commercial poultry experience various conditions that affect their welfare and productivity over their productive lifespan. Laying hens, in particular, may experience stress and fear associated with routine husbandry procedures and interactions with humans [1,2]. Poultry industry transitions go toward cage free housing systems for laying hens where hens are housed in larger groups than in cages, social disruption and injurious pecking among hens may lead to higher levels of stress and detrimental effects on hen welfare [3]. One of the possible alternatives for reducing stress in laying hens is to provide hens with supplementary probiotics.

**Data Availability Statement:** All relevant data are within the paper.

**Funding:** This research was supported by a Fulbright Scholarship (Fulbright Senior Award) to

conduct thermographic examination on birds at Purdue University. Rest of the research on poultry was supported by the USDA National Institute of Food and Agriculture, Foundation program of the Agriculture and Food Research Initiative Competitive Grants Program under the Award No: 2017- 67015-26567. Mention of trade names or commercial products in this article is solely for the purpose of providing specific information and does not imply recommendation or endorsement of the USDA. The USDA is an equal opportunity provider and employer. The funders had no role in study design, data collection and analysis, decision to publish, or preparation of the manuscript.

**Competing interests:** The authors have declared that no competing interests exist.

Recent advances in research have demonstrated the importance of gut microbiota to animal health and behavior [4]. The gut-brain axis is a bidirectional pathway. Gut microbiota influence the central nervous system through neural, endocrine and immune pathways, and thereby influence brain functions in regulating physiological homeostasis and behavior [5,6]. Disturbances in gut microbiota, for example, alter the response of the hypothalamic-pituitary-adrenal (HPA) axis to internal and external stimulation, contributing to various pathogenic conditions such as gut inflammation, reduced gut nutrient absorption, and eating disorders, which adversely affects animal health and increases the costs of production [7].

Probiotics, as nontoxic feed supplements, are gaining momentum because of their beneficial effects on the host's health by improving intestinal microbial balance, increasing nutrient absorption, enhancing local immunity, and reducing gut permeability [8,9]. Studies with humans have demonstrated the roles of probiotic supplementation in decreasing stress responses and improving mood in patients with major depressive disorder [10], irritable bowel syndrome [11] or chronic fatigue [12]. Messaoudi et al. [13], for example, reported beneficial psychological effects and decreased serum cortisol levels with a probiotic formulation (Lactobacillus helveticus R0052 and Bifidobacterium longum R0175) in humans. In rodents, Lactobacillus reduces anxiety, despair-like behaviors and decreases stress-induced plasma corticosterone levels [14]. In farm animals, probiotics have been mostly used to increase production performance and improve animal health and welfare [15]. In particular, probiotics have been used to increase growth performance [16] and to increase resistance to immune challenges [17] and heat stress [18,19] in broiler chickens. Studies with laying hens indicate that probiotics increase feed efficiency, egg production [20] and improve egg quality [21]. Thus, the presence of microorganisms and specific composition of probiotics have critical influences on host physiological and metabolic homeostasis. Consequently, probiotics may be beneficial in reducing stress reactions from various management-related factors under intensified farm animal production systems. However, the potential for using probiotic supplementation to reduce stress reactions of laying hens has not been well investigated.

Stress intensity has been previously measured in laying hens using stress induced hyperthermia (SIH), characterized by an increase in core body temperature together with an initial rapid decrease followed by an increase in skin surface temperature [22–24]. Skin surface temperature can be measured non-invasively using infrared thermography (IRT). IRT detects infrared radiation (heat), providing a pictorial representation of body surface temperature distribution in animals [25], offering both qualitative and quantitative information on the surface temperature of the targeted tissues [26]. IRT in poultry has a broad range of applications including thermoregulation, welfare and stress assessment [27]. Previous research demonstrated that the surface temperature of chickens' head area, including the eye and face regions, varies in response to stress [22,24,28]. Edgar et al. [22] investigated SIH in chickens by measuring the face and eye areas before and after handling procedures. Eye temperature initially decreased by 0.8˚C, then rose to levels significantly higher than baseline temperature. Face temperature also increased over the 20 min post-handling period to reach levels significantly higher than baseline temperature. Similarly, Moe et al. [24] reported that face surface temperature of chickens increased by 0.76˚C in response to handling. The levels of stress and fear that chickens experience are closely related to their genetic background [29]. Compared to commercial brown hens, white hens have higher fear responses [30,31], as indicated by longer durations of tonic immobility (TI).

TI is an anti-predator strategy adopted by various prey animals [32]. Increased TI duration is related to higher fear responses [33,34], and could lead to increased glucocorticoid levels, a common stress indicator [35]. Based on the aforementioned studies, IRT is a useful, non-

invasive tool to detect stress response in laying hens, but stress responses associated with fear have not been examined using IRT.

The overall goal of this study was to assess whether TI-induced stress reactions in laying hens can be reduced by probiotic supplementation and if the temperature alteration, as a stress indicator, is genetically dependent and can been detected with IRT.

## Material and methods

All procedures were approved by the Institutional Animal Care and Use Committee of Purdue University (PACUC number: 1607001454)

### Animals and husbandry

This study was part of a large scale study investigating the effects of probiotic supplementation on the behavior and welfare of laying hens. Hens tested in here had not been handled in the preceding three months.

A total of 396 female chicks of each line, Hy-Line W-36 and Hy-Line Brown, were housed in two rooms (separated by line) at the grower research unit of the Purdue University Animal Sciences Research and Education Center from day old to 12 wk of age. Hens had been randomly distributed to 36 cages per room. A third of the hens from each strain were infrared beak trimmed (BT) at the hatchery as a common practice to reduce aggressive pecking and related injury [36]. Chicks of each line were randomly assigned to three treatments (n = 12 cages/treatment): group G, G'- beak-trimmed chicks fed a regular diet; group B, B'—non-beak trimmed chicks fed the regular diet; and group Y, Y'–non-beak trimmed chicks fed the regular diet supplemented with a probiotic. At 12 wk of age, all pullets were transferred and housed in an enriched caging system at the same research farm. Each cage was equipped with a feed trough, 2 nipple drinkers, perch and nesting area. There were 9 hens/cage, providing 1026 cm2/hen.

The chicks were fed a starter diet containing 20% crude protein, 1.0% Ca, and 0.45% non-phytate phosphorus up to 3.9 wk of age; and a grower diet containing 18.6% crude protein, 1.0% Ca and 0.40% non-phytate phosphorus from 4 to 15.9 wk; then a pre-lay diet with 18.4% crude protein, 2.50% Ca, and 0.35% non-phytate phosphorus from 16 to 17 wk of age, followed by a laying diet with 18.3% crude protein, 4.2% Ca, and 0.3% non-phytate phosphorus. The probiotic diet consisted of the regular diet based on the growth phased mixed with sporulin (Novus International, MO) at 250 ppm. Sporulin contains 3 strains of Bacillus subtillis. Hens had free access to food and water throughout the experiment. The lighting schedule was gradually stepped up to 16L: 8D, which was achieved at 30 wk of age.

### Experimental procedures

In order to investigate the effects of probiotic supplementation on hens' stress reactions, a total of 141 40-wk-old hens, including 71 Hy-Line W-36 hens (group G—24 birds, B—24 birds, and Y- 23 birds) and 70 Brown hens (group G' - 23 birds, B'- 24 birds and Y'- 23birds) were tested in a TI test. Testing was performed in a separate room, adjacent to where hens were housed. TI was performed based on the methods described by Jones and Faure [33]. Each hen was placed on its back in a wooden cradle, with the observer applying slight pressure on the hen's sternum and head for 15 s. A stopwatch was used to record when the bird righted itself. If the bird righted itself in less than 10 s, the restraining procedure was repeated. If TI was not induced after the second attempt, a different bird from the same cage was randomly selected for testing. The maximum duration of TI was 15 min. The ambient temperature in the testing was maintained at 20˚C and the humidity was 70%.

## Infrared thermography

Thermographic images were taken at 2 time points using a thermal camera (uncooled micro-bolometer focal plane array, Focal Plane sensor Array: 640 × 480 pixels, spectral range 7.5–14 μm, accuracy ±1˚C, sensitivity 0.02˚C, InfraTec Dresden, Germany). The first thermographic examination was taken in the testing room immediately before the TI test. For the thermographic examination, each bird was gently held by an observer in a cradle hold with a fixed condition: 0.3 m distance and an 90˚ angle between the thermal imaging camera and the hen's head for all imaging (Fig 1). The emissivity was set to 0.97 for all readings [23]. A second thermographic examination was conducted immediately after the TI test, but before the bird was returned to its original cage, with the entire process of handling and thermographic examination taking less than one min.

## Thermal image analysis

Temperatures were obtained from the thermal images using the software package IRBIS 3 professional software (InfraTec, Dresden, Germany). On each image, each eye temperature (LE—left eye and RE, right eye) was obtained by identifying a spot in the eye center, while maximum

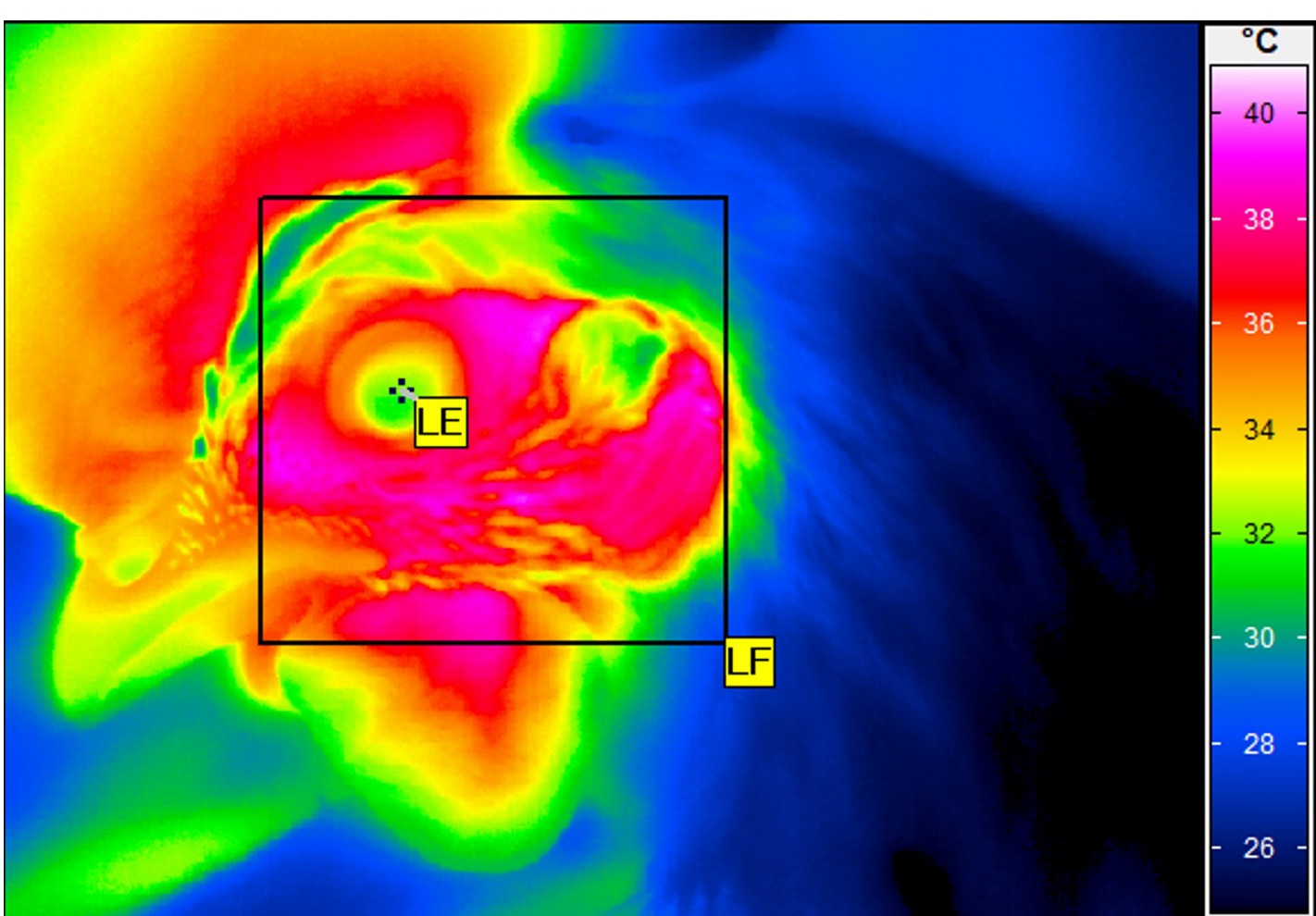

**Fig 1. Thermographic image of hen.** An example of the thermographic image taken from a White Leghorn hen. The image was taken before tonic immobility testing with indicated the measurement regions of the left face (LF) and left eye (LE).

face temperature (LF, left side and RF, right face) was obtained from polygons (including the areas of the face, ear, eye and base of the beak and wattle) (Fig 1), which was similar to the procedures of Edgar et al. [22].

## Statistical analysis

A multivariate analysis of variance (MANOVA) with repeated measures was used for the temperature data. Treatment, time (before and after TI) and genetic line of the hens were considered as fixed effects. In addition, the Spearman rank correlation coefficient was calculated to verify the correlation between the duration of TI and the surface temperature after TI. All the computations were carried out using Statistica 13 software (Dell Inc., Tulsa, OK, USA). Statistically significant differences were reported when $P \leq 0.05$, and a trend was reported when $0.05 < P \leq 0.10$.

## Results

The effect of treatment had no significant influence on the eye and face temperatures ($P = 0.51$; Table 1). The effects of time (before and after TI) and genetic line on eye temperature were significant ($P = 0.03$ and $0.01$, respectively), while the test time by treatment interaction tended toward significance ($P = 0.08$, Table 1).

Both the LE and RE temperatures increased after TI testing ($P < 0.0001$ and $0.03$, respectively, Tables 2 and 3), and the change in RF temperature approached significance ($P = 0.07$, Table 4). White hens had higher average RE and RF temperatures than brown hens ($P = 0.01$, Table 2 and $P = 0.04$, Table 4), and tended to have a higher LE temperature ($P = 0.08$, Table 2). There were no differences in the maximum temperature of the LF (Table 5).

Moderate positive correlations were found between the duration of TI and both eye temperatures after TI in white hens ($rs = 0.36$, $0.34$ and $0.38$ for left, right and average eye temperature, $P < 0.05$, respectively, Table 6).

## Discussion

To our knowledge, the current study is the first to use IRT to examine the effects of probiotic supplementation on reducing stress reactions in laying hens. Previous studies have clearly demonstrated that subjugation to various management stressors, such as changing environmental conditions and diets as well as transportation, disrupt the microenvironment of the gastrointestinal tract in humans and other animals, including chickens [37,38]. Similarly,

**Table 1. The multivariate analysis of variance results for individual factors affecting surface temperature.**

| Effect | Wilks' statistic | F | p |
|---|---|---|---|
| Intercept | 0.00 | 228575.36 | < 0.0001 |
| Line | 0.87 | 3.33 | 0.0137 |
| Treatment | 0.96 | 0.83 | 0.5109 |
| Line×treatment | 0.98 | 0.45 | 0.7710 |
| Test time | 0.89 | 2.80 | 0.0305 |
| Test time×line | 0.98 | 0.55 | 0.6990 |
| Test time×treatment | 0.91 | 2.16 | 0.0800 |
| Test time×line×treatment | 0.93 | 1.57 | 0.1901 |

Test time–before or after TI.

**Table 2. Mean and standard deviations for line, treatment, line and treatment for the temperature of the left eye (LE).**

| Factor | Level | Before TI [˚C] | | After TI [˚C] | | Average (before and after TI) [˚C] | |
|---|---|---|---|---|---|---|---|
| | | n | Mean | n | Mean | n | Mean |
| Line | White | 71 | 32.37±0.56 | 71 | 32.51±0.72 | 142 | 32.44±0.65 |
| | Brown | 70 | 32.09±0.71 | 70 | 32.47±0.66 | 140 | 32.28±0.71 |
| *P*-Value | | | NS | | NS | | *P* = 0.08 |
| Treatment | Group G | 47 | 32.16±0.63 | 47 | 32.47±0.69 | 94 | 32.32±0.68 |
| | Group B | 48 | 32.36±0.67 | 48 | 32.43±0.74 | 96 | 32.40±0.70 |
| | Group Y | 46 | 32.16±0.65 | 46 | 32.57±0.63 | 92 | 32.37±0.67 |
| *P*-Value | | | NS | | NS | | NS |
| Line×treatment | White G | 24 | 32.43±0.61 | 24 | 32.44±0.71 | 48 | 32.43±0.65 |
| | White B | 24 | 32.38±0.51 | 24 | 32.55±0.73 | 48 | 32.47±0.63 |
| | White Y | 23 | 32.31±0.58 | 23 | 32.55±0.73 | 46 | 32.43±0.66 |
| | Brown G' | 23 | 31.88±0.53 | 23 | 32.50±0.69 | 46 | 32.19±0.68 |
| | Brown B' | 24 | 32.35±0.81 | 24 | 32.32±0.75 | 48 | 32.33±0.77 |
| | Brown Y' | 23 | 32.02±0.70 | 23 | 32.59±0.53 | 46 | 32.31±0.68 |
| Average | | 141 | 32.23[a]±0.65 | 141 | 32.49[b]±0.69 | - | *P* < 0.0001 |

[a, b]–different superscript letters denote statistical significance at P≤0.05, TI–tonic immobility, group G, G'—beak trimmed chicks fed a regular diet, group B, B'—non beak trimmed chicks fed the regular diet; group Y, Y'—non beak trimmed chicks fed the regular diet supplemented with probiotics.

physical and or psychological tests including a TI test may cause stress responses in the tested animals.

TI is one of the most common fear tests used in poultry. The process of TI includes capture, handling and manual restraint, which is considered to be stressful to the tested chickens. Increased HPA activation has been evidenced in genetically selected high fear chickens that have a longer average TI duration [39,40]. Longer TI duration has also been revealed in poultry

**Table 3. Means and standard deviations for individual factors for the temperature of the right eye (RE).**

| Factor | Level | Before TI [˚C] | | After TI [˚C] | | Average (before and after TI) [˚C] | |
|---|---|---|---|---|---|---|---|
| | | n | Mean | n | Mean | n | Mean |
| Line | White | 71 | 32.55±0.62 | 71 | 32.65±0.68 | 142 | 32.60[a]±0.65 |
| | Brown | 70 | 32.25±0.67 | 70 | 32.45±0.71 | 140 | 32.35[b]±0.70 |
| *P*-Value | | | NS | | NS | | P = 0.01 |
| Treatment | Group G | 47 | 32.35±0.65 | 47 | 32.47±0.71 | 94 | 32.41±0.68 |
| | Group B | 48 | 32.50±0.69 | 48 | 32.52±0.75 | 96 | 32.51±0.72 |
| | Group Y | 46 | 32.36±0.65 | 46 | 32.66±0.64 | 92 | 32.51±0.66 |
| *P*-Value | | | NS | | NS | | NS |
| Line×treatment | White G | 24 | 32.54±0.68 | 24 | 32.50±0.66 | 48 | 32.52±0.66 |
| | White B | 24 | 32.60±0.56 | 24 | 32.72±0.71 | 48 | 32.66±0.64 |
| | White Y | 23 | 32.51±0.63 | 23 | 32.72±0.69 | 46 | 32.62±0.66 |
| | Brown G' | 23 | 32.15±0.55 | 23 | 32.43±0.77 | 46 | 32.29±0.68 |
| | Brown B' | 24 | 32.39±0.79 | 24 | 32.32±0.76 | 48 | 32.36±0.77 |
| | Brown Y' | 23 | 32.21±0.65 | 23 | 32.59±0.59 | 46 | 32.40±0.64 |
| Average | | 141 | 32.40[a]±0.66 | 141 | 32.55[b]±0.70 | - | P = 0.03 |

[a, b]–different superscript letters denote statistical significance at P ≤ 0.05; TI–tonic immobility, group G, G'—beak trimmed chicks fed a regular diet, group B, B'—non beak trimmed chicks fed a regular diet; group Y, Y'—non beak trimmed chicks fed a diet supplemented with probiotics.

**Table 4. Means and standard deviations for individual factors for the maximum temperature of the right face (RF).**

| Factor | Level | Before TI [°C] | | After TI [°C] | | Average (before and after TI) [°C] | |
|---|---|---|---|---|---|---|---|
| | | n | Mean | n | Mean | n | Mean |
| Line | White | 71 | 39.47±0.43 | 71 | 39.56±0.43 | 142 | 39.51[a]±0.43 |
| | Brown | 70 | 39.34±0.56 | 70 | 39.39±0.54 | 140 | 39.306[b]±0.55 |
| P-Value | | | NS | | NS | | P = 0.04 |
| Treatment | Group G | 47 | 39.45±0.48 | 47 | 39.57±0.49 | 94 | 39.51±0.49 |
| | Group B | 48 | 39.39±0.55 | 48 | 39.49±0.48 | 96 | 39.44±0.52 |
| | Group Y | 46 | 39.37±0.46 | 46 | 39.37±0.50 | 92 | 39.37±0.48 |
| P-Value | | | NS | | NS | | NS |
| Line×treatment | White G | 24 | 39.52±0.45 | 24 | 39.59±0.40 | 48 | 39.55±0.42 |
| | White B | 24 | 39.36±0.47 | 24 | 39.61±0.44 | 48 | 39.49±0.47 |
| | White Y | 23 | 39.52±0.34 | 23 | 39.48±0.46 | 46 | 39.50±0.40 |
| | Brown G' | 23 | 39.37±0.51 | 23 | 39.55±0.58 | 46 | 39.46±0.55 |
| | Brown B' | 24 | 39.41±0.63 | 24 | 39.37±0.50 | 48 | 39.39±0.57 |
| | Brown Y' | 23 | 39.23±0.53 | 23 | 39.25±0.53 | 46 | 39.24±0.52 |
| Average | | 141 | 39.40±0.50 | 141 | 39.48±0.49 | - | P = 0.07 |

[a, b]–different superscript letters denote statistical significance at P ≤ 0.05, TI–tonic immobility, group G, G'—beak trimmed chicks fed a regular diet, group B, B'—non beak trimmed chicks fed a regular diet; group Y, Y'—non beak trimmed chicks fed a diet supplemented with probiotics.

exposed to other non-handling stressors, such as heat stress [41] and dietary mycotoxin [42]. Stress upregulates the HPA axis, leading to the release of corticotropin releasing factor and cortisol [43–45], and has a direct effect on the gastrointestinal tract to increase intestinal permeability, known as "leaky gut" [46]. Probiotics, as beneficial bacteria, have been used as therapeutics for stress related inflammatory bowel disease [47,48] and mental disorders in humans [49]. The underlying cellular mechanisms of probiotic functions are able to inhibit binding of

**Table 5. Means and standard deviations for individual factors for the maximum temperature of the left face (LF).**

| Factor | Level | Before TI [°C] | | After TI [°C] | | Average (before and after TI) [°C] | |
|---|---|---|---|---|---|---|---|
| | | n | Mean | n | Mean | n | Mean |
| Line | White | 71 | 39.53±0.42 | 71 | 39.56±0.42 | 142 | 39.54±0.42 |
| | Brown | 70 | 39.49±0.51 | 70 | 39.56±0.51 | 140 | 39.52±0.51 |
| P-Value | | | NS | | NS | | NS |
| Treatment | Group G | 47 | 39.58±0.47 | 47 | 39.64±0.43 | 94 | 39.61±0.45 |
| | Group B | 48 | 39.53±0.47 | 48 | 39.58±0.44 | 96 | 39.56±0.45 |
| | Group Y | 46 | 39.40±0.45 | 46 | 39.45±0.51 | 92 | 39.42±0.48 |
| P Value | | | NS | | NS | | NS |
| Line× treatment | White G | 24 | 39.67±0.48 | 24 | 39.61±0.47 | 48 | 39.64±0.47 |
| | White B | 24 | 39.47±0.38 | 24 | 39.59±0.37 | 48 | 39.53±0.38 |
| | White Y | 23 | 39.44±0.36 | 23 | 39.47±0.42 | 46 | 39.45±0.38 |
| | Brown G' | 23 | 39.50±0.45 | 23 | 39.68±0.40 | 46 | 39.59±0.43 |
| | Brown B' | 24 | 39.60±0.55 | 24 | 39.58±0.50 | 48 | 39.59±0.52 |
| | Brown Y' | 23 | 39.36±0.53 | 23 | 39.42±0.60 | 46 | 39.39±0.56 |
| Average | | 141 | 39.51±0.47 | 141 | 39.56±0.47 | - | NS |

TI–tonic immobility, group G, G'—beak trimmed chicks fed a regular diet, group B, B'—non beak trimmed chicks fed the regular diet; group Y, Y'—non beak trimmed chicks fed the regular diet supplemented with probiotics.

**Table 6. The Spearman rank correlation coefficients between the time of tonic immobility and surface temperature after tonic immobility for White Leghorn and Hy-Line brown line (n = 141).**

| Variable | $r_s$ |
|---|---|
| White | |
| LE temperature | 0.36* |
| RE temperature | 0.34* |
| AE temperature | 0.38* |
| Brown | |
| LE temperature | -0.08 |
| RE temperature | -0.04 |
| AE temperature | -0.03 |

LE–left eye, RE–right eye, AE–average eye temperature of the left and right eye.

pathogenic bacteria to intestinal epithelial cells, enhance barrier function and suppress the growth of pathogens [50]. The intestinal microbiota also have an influence on the synthesis and release of various neuroactive factors, neurotransmitters and neuromodulators, by which microbiota directly and indirectly deliver signals to the brain, thereby adjusting stress reactions via the gut-brain axis [43–45].

In poultry, dietary supplementation of probiotics has been used to reduce deleterious effects of heat stress [18, 19]. However, the results from the current experiment did not support our hypothesis that probiotics supplementation can provide a modulatory effect on TI-induced stress in laying hens. A possible explanation for this is that compared to other stressors such as heat stress, TI is a relatively mild and acute stressor which might not be enough to cause gastrointestinal disruption. In addition, physiological parameters, including heart rate, body temperature and brain activities (measured by electroencephalogram), are usually elevated immediately when TI is induced and gradually return to baseline before TI is terminated [51]. Probiotics may have potential to attenuate the stress response if the animals are exposed to a more prolonged or more intense stressor.

Although there was a lack of treatment difference, the left and right eye temperatures were increased by 0.26°C and 0.15°C, respectively, following TI testing. This finding is in agreement with other studies on stress reactions in animals, including poultry. Stress-induced hyperthermia has been characterized by increased core temperature and decreased surface temperature as a part of the broad physiological process of body hyperthermia [52]. Previous research has determined that core body temperature can be evaluated noninvasively using eye temperature measurements in cattle [53], and horses [54]. In laying hens, core body temperature initially decreased following the induction of TI, while hyperthermia occurred shortly after the response terminated [51,55,56]. In the current study, core temperature was not measured to avoid adding further stress to the birds. In addition, due to the restriction of group housing, we were unable to take baseline measurements of individual hens' face and eye to indicate the temperature changes in response to the initial stress stimuli (catching, removing, and handling); instead, the comparison was made between before and after TI testing for each bird. However, the current results indicate that the increased eye temperature could be used as a stress indicator of chickens in response to TI. Similarly, eye temperature change in response to stressors has been reported in various poultry studies. Edgar et al. [22] reported that eye temperature immediately decreased, reaching a minimum of 0.8°C within the first 3 minutes and then increased significantly after 10 minutes of handling. In a recent study, hens that wore an unfamiliar device for behavior monitoring had higher eye temperatures in the acclimation

period compared to hens in the control group [57]. In addition, Herborn et al. [58] reported that face and eye surface temperature increases could also be used as a long-term marker for stress responses in laying hens. In contrast, Herborn et al. [23] reported an initial drop in eye temperature of 0.4˚C in reaction to handling only, with no temperature elevation mentioned. In our study, the average duration of TI was 7.2 minutes and we were still able to detect a temperature increase in both eyes, which indicate SIH could be a useful non-invasive method for stress detection, even if the stressor is considered to be relatively mild or short-lasting.

Interestingly, temperature only tended to increase in the right side of the face following TI testing, which partially agrees with results reported by Edgar et al. [22] and Moe et al. [24]. Edgar et al. [22] reported that 20 minutes was needed to monitor temperature changes in the head region in response to a handling procedure. In the present study, the maximum time allotted for TI was 15 minutes, with an average duration of 7.2 min. Shorter duration of TI could be a reason why face temperature tended to increase only on one side. Possible longer TI would significantly increase face temperature and decrease the level of discrepancy between both sides of the face, as according to Jones and Mills [59] a longer TI response is related to increased fearfulness and therefore stress reactions.

In farm animal industries, genetic traits play an important role in numerous areas, such as production yield, stress response, immune function, skeletal health, and behavior patterns. Our results indicate that genetic line has a significant influence on temperatures in the head areas. The average temperature before and after TI of the white hens was warmer in RE and RF, with a tendency for warmer temperatures in LE compared to brown hens. The results are in agreement with previous studies indicating that white hens are more fearful and have higher stress reactions than brown hens [29, 30, 33, 60]. The genetic difference could also explain the current findings of the lack of correlation between the surface temperature changes and TI duration in brown hens, whereas white hens had a moderate positive correlation between eye temperature change and TI duration, as well as the temperatures of the LE and RE. In other words, there is a higher temperature increase with longer TI duration. These findings agree with the results presented by Edgar [22], where eye temperature rose gradually as the handling time increased. The current results are also supported by studies showing that the birds selected for long TI duration have upregulated stress responses compared to birds with a shorter TI duration [39, 61].

## Conclusions

The study is the first to demonstrate the effects of probiotic supplementation on the stress response of laying hens assessed by IRT. Both LE and RE temperatures were significantly increased in response to TI testing, indicating a stress response. There was a correlation between the duration of TI and the TI-induced temperature changes of the LE and RE for white hens. In addition, white hens had higher temperatures on the right side of the head, suggesting a greater reaction to TI than brown hens. However, probiotic supplementation did not affect eye and face temperatures measured by IRT. These results suggest that IRT could be used as a non-invasive technique to assess stress, especially in white hens.

## Acknowledgments

We would like to thank the staff and graduate students of the Livestock Behavior Research Unit, USDA-ARS and the Department of Animal Sciences at Purdue University.

## Author Contributions

**Conceptualization:** Maria Soroko.

**Formal analysis:** Daniel Zaborski.

**Investigation:** Maria Soroko.

**Methodology:** Maria Soroko.

**Supervision:** Daniel Zaborski.

**Writing – original draft:** Maria Soroko, Daniel Zaborski.

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
