## [Decision Letter · Decision Letter 0]

8 May 2020

PONE-D-20-10311

Investigation of the effects of probiotic, Bacillus subtilis on stress reactions in
laying hens using infrared thermography

PLOS ONE

Dear dr Soroko,

Thank you for submitting your manuscript to PLOS ONE. After careful consideration, we
feel that it has merit but does not fully meet PLOS ONE’s publication criteria as it
currently stands. Therefore, we invite you to submit a revised version of the
manuscript that addresses the points raised during the review process.

One
reviewer made some minor editorial suggestions/comments that will help clarify the
manuscript.  Please respond specifically to these comments in your
revision.

We would appreciate receiving your revised manuscript by Jun 22 2020 11:59PM. When
you are ready to submit your revision, log on to https://www.editorialmanager.com/pone/ and select the 'Submissions
Needing Revision' folder to locate your manuscript file.

If you would like to make changes to your financial disclosure, please include your
updated statement in your cover letter.

To enhance the reproducibility of your results, we recommend that if applicable you
deposit your laboratory protocols in protocols.io, where a protocol can be assigned
its own identifier (DOI) such that it can be cited independently in the future. For
instructions see: http://journals.plos.org/plosone/s/submission-guidelines#loc-laboratory-protocols

We look forward to receiving your revised manuscript.

Kind regards,

Michael H. Kogut, Ph.D.

Academic Editor

PLOS ONE

Journal Requirements:

"No"

"No"

a. Please complete your Competing Interests statement to state any Competing
Interests. If you have no competing interests, please state "The authors have
declared that no competing interests exist.", as detailed online in our guide for
authors at http://journals.plos.org/plosone/s/submit-now

Reviewers' comments:

Reviewer's Responses to Questions

**Comments to the Author**

1. Is the manuscript technically sound, and do the data support the conclusions?

Reviewer #1: Yes

Reviewer #2: Yes

2. Has the statistical analysis been performed
appropriately and rigorously? 

Reviewer #1: Yes

Reviewer #2: Yes

3. Have the authors made all data underlying the
findings in their manuscript fully available?

Reviewer #1: Yes

Reviewer #2: Yes

4. Is the manuscript presented in an intelligible
fashion and written in standard English?

Reviewer #1: Yes

Reviewer #2: Yes

5. Review Comments to the Author

Reviewer #1: I have not found significant limitations; conversely, I think that this
manuscript has many strengths, such as originality and methodological accuracy. I
think that this paper I reviewed reports novel findings useful for poultry health
and wellbeing. The methods, results and data interpretation meet the journal's
standard; no errors have been detected and the conclusions have been properly
supported. The trials were well described and the findings have been well discussed.
Moreover, the language is also fine.

Reviewer #2: The manuscript investigates the use of a probiotic in laying hens as a
way to reduce stress. To evaluate this stress, infrared thermography was used. The
manuscript uses good approaches and is acceptable with regard to content. The
manuscript concludes that probiotics did not reduce stress under the conditions
evaluated. However, infrared thermography could be used to evaluate stress in laying
hens.

Page 2, line 24-27: Why was a beak trimmed group included in the study? Should there
have been a beak trimmed with probiotics to effectively compare?

The abstract reads that ”Seventy-one white and 70 brown hens were used.” But page 6,
line 109 reads, “396 female chicks of each line” were housed. I assume this is the
starting number of conditioned/pullets that will be selected from for the study.

To clarify the materials and methods, 141 birds were broken into 3 groups which will
break the groups into 47 birds each. This would be further broken down into 23-24
Hyline W-36 hens and 23-24 Brown hens per group.

Page 10, line 171-173: Results. The beak trimmed birds and non-beaked trimmed were
pooled for further analysis. How were these pooled with the other groups? Were they
included in the non-probiotic groups or added to both of the other groups? Please
clarify.

Is there a significance for the references that were in red or were these
corrections?

6. PLOS authors have the option to publish the peer
review history of their article (what does this mean?). If published, this will
include your full peer review and any attached files.

If you choose “no”, your identity will remain anonymous but your review may still be
made public.

**Do you want your identity to be public for this peer review?** For
information about this choice, including consent withdrawal, please see our
Privacy Policy.

Reviewer #1: Yes: Guillermo Tellez-Isaias

Reviewer #2: Yes: James A Byrd

---

## [Author Response · Author response to Decision Letter 0]

18 May 2020

Responses to reviewers' comments, manuscript Ref: PONE-D-20-10311, entitled
“Investigation of the effects of probiotic, Bacillus subtilis on stress reactions in
laying hens using infrared thermography”.

Comments to the Author

1. Comment: Please ensure that your manuscript meets PLOS ONE's style requirements,
including those for file naming. The PLOS ONE style templates can be found at:

Response: The article style according to journal requirements has been checked and
corrected. 

2. Comment: Thank you for stating the following financial disclosure:

"No"

a. Please clarify the sources of funding (financial or material support) for your
study. List the grants or organizations that supported your study, including funding
received from your institution.

Response: The authors received no specific funding for this work. Maria Soroko was
supported by Fulbright Scholarship (Fulbright Senior Award) to conduct thermographic
examination on birds at Purdue University. Rest of the research on poultry was
supported by the USDA National Institute of Food and Agriculture, Foundation program
of the Agriculture and Food Research Initiative Competitive Grants Program under the
Award No: 2017-67015-26567. Mention of trade names or commercial products in this
article is solely for the purpose of providing specific information and does not
imply recommendation or endorsement of the USDA. The USDA is an equal opportunity
provider and employer.

Response: The funders had no role in study design, data collection and analysis,
decision to publish, or preparation of the manuscript.

Response: None of the authors received any financial support. 

d. If you did not receive any funding for this study, please state: “The authors
received no specific funding for this work.”

Response: The authors received no specific funding for this work.

Response: Amended statements have been added in the cover letter. 

3. Comment: Thank you for stating the following in your Competing Interests section: 

"No"

a) Please complete your Competing Interests statement to state any Competing
Interests. If you have no competing interests, please state "The authors have
declared that no competing interests exist.", as detailed online in our guide for
authors at http://journals.plos.org/plosone/s/submit-now

Response: The authors have declared that no competing interests exist. The
information has been included in the cover letter.

Reviewers' comments:

Is the manuscript technically sound, and do the data support the conclusions?

Reviewer #1: Yes

Reviewer #2: Yes

Has the statistical analysis been performed appropriately and rigorously?

Reviewer #1: Yes

Reviewer #2: Yes

Have the authors made all data underlying the findings in their manuscript fully
available?

Reviewer #1: Yes

Reviewer #2: Yes

Is the manuscript presented in an intelligible fashion and written in standard
English?

Reviewer #1: Yes

Reviewer #2: Yes

Review Comments to the Author

Reviewer #1: I have not found significant limitations; conversely, I think that this
manuscript has many strengths, such as originality and methodological accuracy. I
think that this paper I reviewed reports novel findings useful for poultry health
and wellbeing. The methods, results and data interpretation meet the journal's
standard; no errors have been detected and the conclusions have been properly
supported. The trials were well described and the findings have been well discussed.
Moreover, the language is also fine.

Reviewer #2: The manuscript investigates the use of a probiotic in laying hens as a
way to reduce stress. To evaluate this stress, infrared thermography was used. The
manuscript uses good approaches and is acceptable with regard to content. The
manuscript concludes that probiotics did not reduce stress under the conditions
evaluated. However, infrared thermography could be used to evaluate stress in laying
hens.

1. Comment: Page 2, line 24-27: Why was a beak trimmed group included in the study?
Should there have been a beak trimmed with probiotics to effectively compare?

Response: Thank you for your comment. We included beak trimmed group fed regular diet
because according to previous studies beak trimming reduces stress response in
birds. Therefore we wanted to indicate if there are any temperature changes between
beak trimmed – fed regular diet, non-beak trimmed fed a regular diet; and non-beak
trimmed fed a diet supplemented with probiotics. Future research could include
temperature differences between: beak trimmed fed a diet supplemented with
probiotics and beak trimmed fed a regular diet (as a control group).

2. Comment: The abstract reads that ”Seventy-one white and 70 brown hens were used.”
But page 6, line 109 reads, “396 female chicks of each line” were housed. I assume
this is the starting number of conditioned/pullets that will be selected from for
the study.

Response: Thank you for your comment. I confirm 396 female chicks of each line that
was a starting number and in the research we included a part od those birds n = 141
of both lines. 

3. Comment: To clarify the materials and methods, 141 birds were broken into 3 groups
which will break the groups into 47 birds each. This would be further broken down
into 23-24 Hyline W-36 hens and 23-24 Brown hens per group.

Response: Thank you for that comment: 141 bird included 2 lines: 71 Hy-Line W-36 hens
and 70 Brown hens. Each line was broken into 3 groups: including 71 Hy-Line W-36
hens (group G - 24 birds, B - 24 birds, and Y- 23 birds) and 70 Brown hens (group G’
- 23 birds, B’- 24 birds and Y’- 23 birds). More information about groups has been
added at lines: 132-133 page 7 and 8. 

4. Comment: Page 10, line 171-173: Results. The beak trimmed birds and non-beaked
trimmed were pooled for further analysis. How were these pooled with the other
groups? Were they included in the non-probiotic groups or added to both of the other
groups? Please clarify.

Response: Thank you for very much your comment. This sentence is superfluous. It has
been delete from the manuscript. 

5. Comment: Is there a significance for the references that were in red or were these
corrections?

Response: Thank you for pointing this. There was no reason for highlighting
references, it was done by mistake.

to Reviewers.docx
---

## [Editor Report · Decision Letter 1]

20 May 2020

Investigation of the effects of probiotic, Bacillus subtilis on stress reactions in
laying hens using infrared thermography

PONE-D-20-10311R1

Dear Dr. Górniak,

We are pleased to inform you that your manuscript has been judged scientifically
suitable for publication and will be formally accepted for publication once it
complies with all outstanding technical requirements.

With kind regards,

Michael H. Kogut, Ph.D.

Academic Editor

PLOS ONE
---

## [Editor Report · Acceptance letter]

2 Jun 2020

PONE-D-20-10311R1 

Investigation of the effects of probiotic, Bacillus subtilis on stress reactions in
laying hens using infrared thermography 

Dear Dr. Soroko:

I'm pleased to inform you that your manuscript has been deemed suitable for
publication in PLOS ONE. Congratulations! Your manuscript is now with our production
department. 

Kind regards, 

on behalf of

Dr. Michael H. Kogut 

Academic Editor

PLOS ONE